# A Transient Event-Capturing Circuit and Adaptive PI Control for a Voltage Mode Superbuck Converter

Yinyu Wang [1] , Baoqiang Huang [1], Yuanxun Wang [2], Haoran Xu [1], Desheng Zhang [3],* and Qiaoling Tong [1]

1 School of Integrated Circuits, Huazhong University of Science and Technology, Wuhan 430074, China; yingyuwang@hust.edu.cn (Y.W.); d202387049@hust.edu.cn (B.H.); m202272786@hust.edu.cn (H.X.); tongqiaoling@hust.edu.cn (Q.T.)
2 School of International Education, Wuhan University of Technology, Wuhan 430074, China; wangyxun@whut.edu.cn
3 School of Automation, Wuhan University of Technology, Wuhan 430074, China
* Correspondence: dszhang@whut.edu.cn

**Abstract:** This paper proposes a transient event-capturing circuit and adaptive PI control to monitor and improve the transient response of a superbuck converter. The transient event-capturing circuit is composed of coupling and capturing circuits. The coupling circuit converts the output voltage to the sensed voltage, whereas the DC and ripple components are eliminated. By counting the up-crossing and down-crossing numbers of the sensed voltage, the capturing circuit classifies the output voltage response into different transient events according to oscillation cycles. The transient events carry the stability information that can be used to adjust the bandwidth and phase margin. Finally, an adaptive PI controller is implemented with the proposed transient event-capturing circuit to improve the stability and transient response. Experimental results of the 100 W superbuck converter verify the effectiveness of the adaptive PI controller for improving the transient response and stability. The adaptive PI controller eliminates the oscillations due to deviated parameters and operating conditions. The maximum oscillation amplitude is reduced from 2 V to 400 mV at the reference voltage change.

**Keywords:** capture; DC–DC; feedback control; monitor; superbuck; stability; transient event; transient response

## 1. Introduction

A superbuck converter is a derivative of Cuk/Sepic converters with a canonical switching cell, which is suitable for photovoltaic and aerospace applications owing to the low EMI noise and continuous input and output currents [1–3], as shown in Figure 1. Its dynamic behavior is complicated due to the fourth-order system characteristic. A small signal model of the superbuck converter has four poles and two right half-plane (RHP) zeros. Therefore, transient response optimization is a huge challenge for designing superbuck converters.

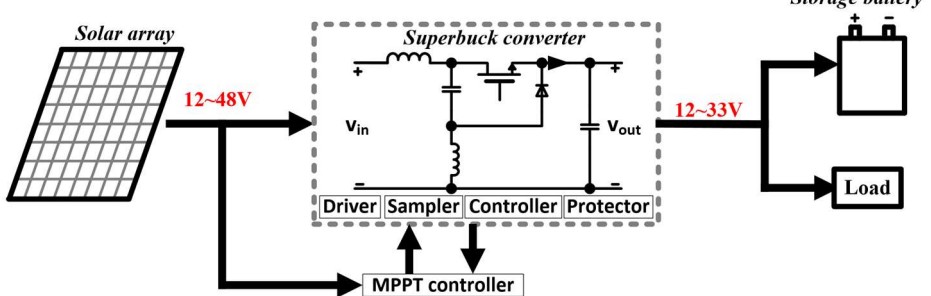

**Figure 1.** Superbuck converter applied in solar power generation of photovoltaics.

To simplify the stability improvement of a superbuck, RHP zeros are moved to the left half plane (LHP) by the damping network [4]. However, this approach may fail due to deviated parameters and operation conditions. Considering two RHP zeros, sliding mode control (SMC) can improve the response speed, although it would need four samplings [5,6]. To simplify the control, hysteresis SMC control with a novel sliding surface can reduce the sampling required [7]. To improve the transient response, the predicted peak current mode (PPCM) controls strictly regulate the inductance current [8–11]. With PPCM control, the RHP zeros are eliminated when the two inductor values satisfy a certain relationship [12]. However, the above control methods cannot be adapted to the wide operating range.

Monitoring and adaptive controls provide novel solutions to improve the stability of deviated parameters and wide operating ranges. For model deviations with a square-wave injected into the feedback loop, an online stability margin monitor injects the square-wave into the feedback loop to acquire and adjust the phase margin [13–15]. Furthermore, for a wide operating range, based on phase margin monitoring, an adaptive phase margin controller with proportional–integral (PI) parameters is proposed [16–18]. The above controls dynamically extend the bandwidth while maintaining the phase margin under different operating conditions.

While the phase margin monitoring needs to inject the small signal in the control loop, disturbance observation monitors the output voltage directly, which eliminates the small signal injection [19,20]. Based on disturbance observation, researchers have proposed disturbance-observer-based feedback control [21,22], integral sliding mode controls [23,24], etc. In these approaches, the observed signals have to be processed by the main controller, which require costly hardware for implementation. To reduce communications and calculations, the observed results can be processed in the sensing network. Furthermore, event-triggered adaptive controls have been used for different nonlinear and uncertain systems [25–27]. However, relevant studies are lacking for a superbuck converter with a nonlinear fourth-order model.

To address the above-mentioned issues, this paper proposes a transient event-capturing circuit and adaptive PI control to monitor and improve the transient response of a superbuck converter. The coupling circuit converts the output voltage to the sensed voltage, whereas the DC and ripple components are eliminated. By counting the up-crossing and down-crossing numbers of the sensed voltage, the capturing circuit classifies the output voltage response into different transient events according to oscillation cycles. The transient events carry the stability information, which can be used to adjust the bandwidth and phase margin. Finally, an adaptive PI controller is carried out with the proposed transient event-capturing circuit to improve the stability and transient response. The gain of the adaptive PI controller is designed according to the weighting factors.

This paper is organized as follows. Section 2 provides small signal models and stability analysis for the superbuck converter. In Section 3, the transient event coupling and capturing circuits are presented along with analyses of potential components in the converter output. An adaptive PI controller is designed with the captured transient events. The effectiveness of the proposed strategy is verified by simulations and experiments in Sections 4 and 5. Finally, a brief conclusion is provided in Section 6.

## 2. Small-Signal Modeling and Stability of the Superbuck Converter

A basic superbuck converter is shown in Figure 2. Conventionally, a damping network composed of $R_d$ and $C_d$ is used to compensate the RHP zeros.

Considering voltage-second balancing of inductors and charge balancing of capacitors, the steady state operation equations are given by:

$$\begin{cases} V_{out} = DV_{in} \\ V_{C1} = V_{in} \\ I_{L1} = \frac{DV_{out}}{R} \\ I_{L2} = \frac{(1-D)V_{out}}{R} \end{cases}, \tag{1}$$

where, $V_{in}$, $V_{out}$, $V_{C1}$, $I_{L1}$ and $I_{L2}$ are DC values of input voltage, output voltage, the capacitance voltage of $C_1$, inductance currents of $L_1$ and $L_2$. The duty cycle, $D$, is the ratio between the time of $Q_1$ on and $Q_2$ off and the switching period.

As (1) indicates, the output voltage at steady state is identical to conventional buck converter. However, the dynamic behavior is more complicated, and the small signal model is a fourth-order system.

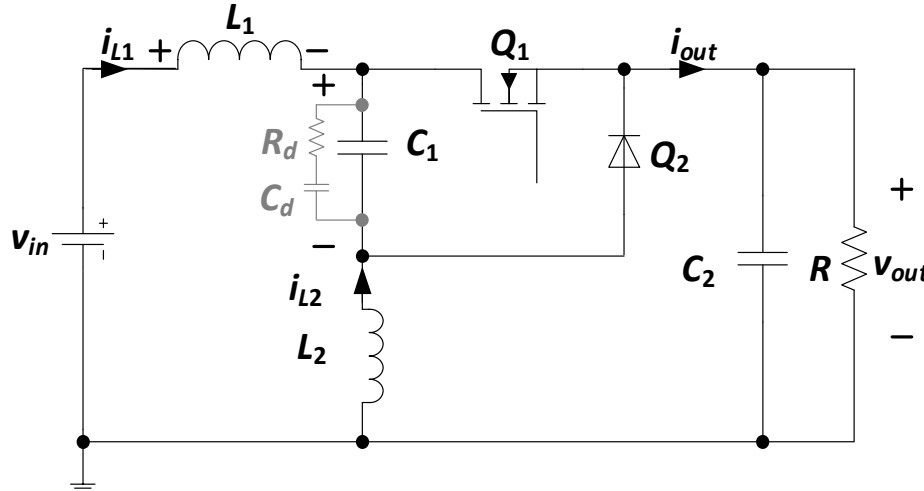

**Figure 2.** Superbuck converter scheme.

### 2.1. Small Signal Model of the Superbuck Converter

According to Figure 2, there are two states of the superbuck converter without the damping network, as shown in Figure 3a,b.

According to Figure 3a, at $Q_1$ on $Q_2$ off, the current and voltage of $L_1$ and $L_2$, and the current and voltage of $C_1$ and $C_2$ satisfy

$$
\begin{cases}
L_1 \frac{di_{L1}}{dt} = v_{in} - v_{out} \\
L_2 \frac{di_{L2}}{dt} = -(v_{out} - v_{C1}) \\
C_1 \frac{dv_{C1}}{dt} = -i_{L2} \\
C_2 \frac{dv_{C2}}{dt} = i_{L1} + i_{L2} - \frac{v_{out}}{R}
\end{cases}. \tag{2}
$$

According to Figure 3b, at $Q_1$ off $Q_2$ on, the current and voltage of $L_1$ and $L_2$, and the current and voltage of $C_1$ and $C_2$ satisfy

$$
\begin{cases}
L_1 \frac{di_{L1}}{dt} = v_{in} - v_{C1} - v_{out} \\
L_2 \frac{di_{L2}}{dt} = -v_{out} \\
C_1 \frac{dv_{C1}}{dt} = i_{L1} \\
C_2 \frac{dv_{C2}}{dt} = i_{L1} + i_{L2} - \frac{v_{out}}{R}
\end{cases}, \tag{3}
$$

Therefore, combing (2) and (3), the average state-space model is derived as

$$
\begin{cases}
L_1 \frac{di_{L1}}{dt} = v_{in} - v_{out} - v_{C1}(1-d) \\
L_2 \frac{di_{L2}}{dt} = -v_{out} + v_{C1}d \\
C_1 \frac{dv_{C1}}{dt} = -i_{L2}d + i_{L1}(1-d) \\
C_2 \frac{dv_{out}}{dt} = i_{L1} + i_{L2} - \frac{v_{out}}{R}
\end{cases}, \tag{4}
$$

where $v_{in}$, $v_{out}$, $v_{C1}$, $i_{L1}$, and $i_{L2}$ are period average values of input voltage, output voltage, capacitance voltage of $C_1$, and inductance currents of $L_1$ and $L_2$. $d$ is the period average value of duty cycle.

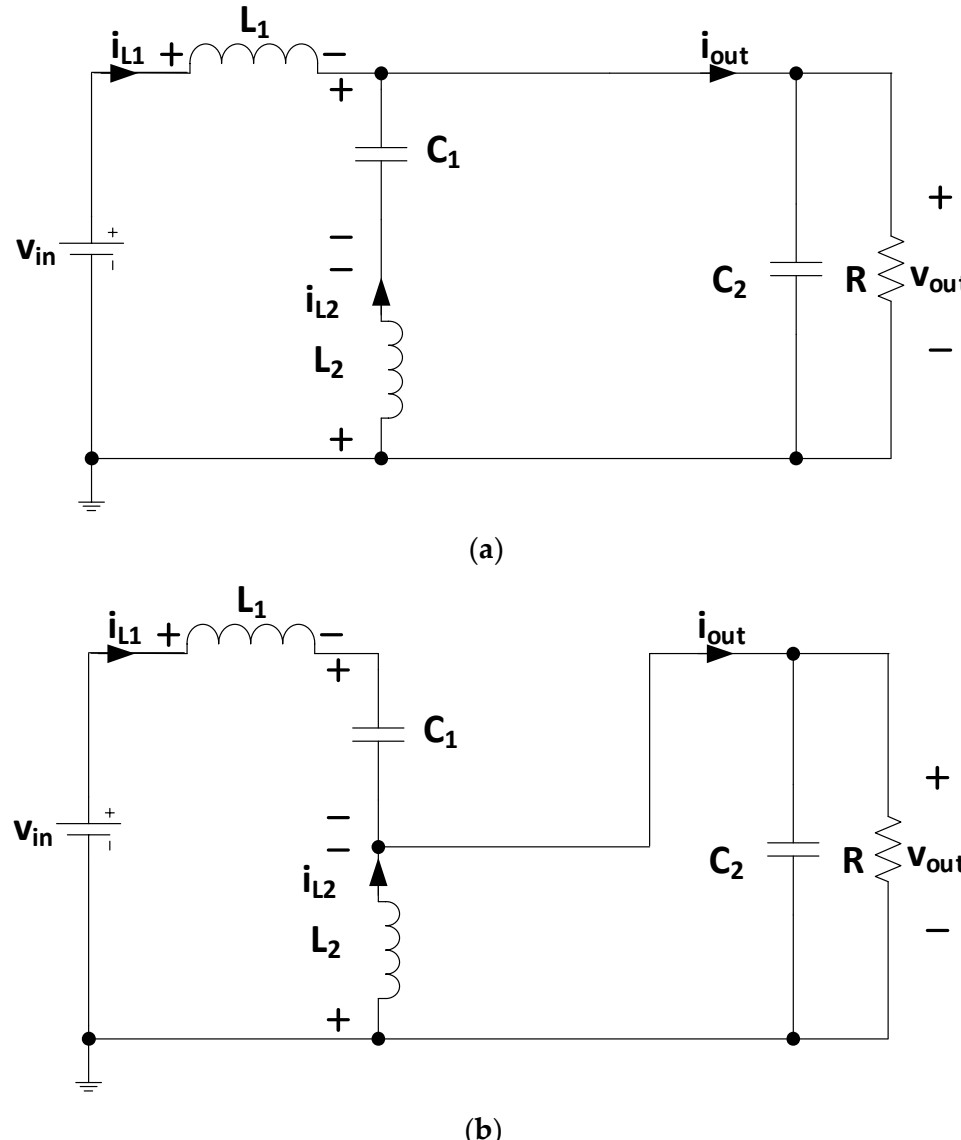

**Figure 3.** Two states of the superbuck converter: (**a**) $Q_1$ on, $Q_2$ off; (**b**) $Q_1$ off, $Q_2$ on.

According to (4), the small signal model is derived as

$$
\begin{cases}
L_1 \frac{d\hat{i}_{L1}}{dt} = \hat{v}_{in} - \hat{v}_{out} - \hat{v}_{C1}(1-D) + v_{C1}\hat{d} \\
L_2 \frac{d\hat{i}_{L2}}{dt} = -\hat{v}_{out} + \hat{v}_{C1}D + v_{C1}\hat{d} \\
C_1 \frac{d\hat{v}_{C1}}{dt} = -\hat{i}_{L2}D + \hat{i}_{L1}(1-D) - i_{L2}\hat{d} - i_{L1}\hat{d} \\
C_2 \frac{d\hat{v}_{out}}{dt} = \hat{i}_{L1} + \hat{i}_{L2} - \frac{\hat{v}_{out}}{R} + \frac{v_{out}}{R^2}\hat{R}
\end{cases} \tag{5}
$$

where $\hat{v}_{in}$, $\hat{v}_{out}$, $\hat{v}_{C1}$, $\hat{i}_{L1}$, $\hat{i}_{L2}$ and $\hat{d}$ are small signals of $v_{in}$, $v_{out}$, $v_{C1}$, $i_{L1}$, $i_{L2}$ and $d$.

Furthermore, from (5), the small signal model in Laplace domain is given by

$$
\begin{cases}
s\hat{i}_{L1} = \frac{1}{L_1}[\hat{v}_{in} - \hat{v}_{out} - \hat{v}_{C1}(1-D) + v_{C1}\hat{D}] \\
s\hat{i}_{L2} = \frac{1}{L_2}[-\hat{v}_{out} + \hat{v}_{C1}D + v_{C1}\hat{D}] \\
s\hat{v}_{C1} = \frac{1}{C_1}[-\hat{i}_{L2}D + \hat{i}_{L1}(1-D) - i_{L2}\hat{D} - i_{L1}\hat{D}] \\
s\hat{v}_{out} = \frac{1}{C_2}[\hat{i}_{L1} + \hat{i}_{L2} - \frac{\hat{v}_{out}}{R} + \frac{v_{out}}{R^2}\hat{R}]
\end{cases} \tag{6}
$$

Based on (6), the transfer function between the output voltage and the duty cycle is derived as

$$G_{vd}(s) = \frac{\hat{v}_{out}}{\hat{d}} = V_{in}\frac{s^2(L_1+L_2)C_1R - sDa + R}{s^4L_1L_2C_1C_2 + s^3L_1C_1L_2 + s^2[C_2b + (L_1+L_2)C_1]R + sb + R}, \quad (7)$$

where $a$ and $b$ are given by $a = DL_1 - (1-D)L_2$ and $b = D^2L_1 + (1-D)^2L_2$.

In (7), there are two zeros $\omega_{z1}$ and $\omega_{z2}$ in the transfer function, which are given by

$$\omega_{z1}, \omega_{z2} = \frac{\frac{Da}{R} \pm \sqrt{\left(\frac{Da}{R}\right)^2 - 4(L_1+L_2)C_1}}{2(L_1+L_2)C_1}. \quad (8)$$

Therefore, when $a$ satisfies $a > 0$, two zeros are located at the RHP.

Considering the damping network, the transfer function is revised as:

$$G_{vd}(s) = \frac{V_{in}\{s^2R(L_1+L_2)c(s) - sDa + R\}}{s^4L_1L_2RC_2c(s) + s^3L_1L_2c(s) + s^2R[C_2b + (L_1+L_2)c(s)] + sb + R}, \quad (9)$$

where $c(s)$ is given by $c(s) = (sR_dC_dC_1 + C_1 + C_d)/(sR_dC_d + 1)$. When $\omega >> 1/R_dC_d$ is valid, $c(s)$ is approximate to $C_1 + 1/sR_d$ and (9) becomes:

$$G_{vd}(s) = \frac{v_{in}\{s^2(L_1+L_2)RC_1 + s[-aD + (L_1+L_2)R/R_d] + R\}}{s^4L_1L_2RC_1C_2 + s^3L_1L_2(C_1 + RC_2/R_d) + s^2[RC_2b + (L_1+L_2)RC_1 + L_1L_2/R_d] + s[(L_1+L_2)R/R_d + b] + R}. \quad (10)$$

As (10) indicates, the original RHP zeros can be moved to the LHP under the condition of $(L_1 + L_2)R/R_d > aD$. This greatly improves the stability, especially when the LHP zeros are near the dominant poles. When the LHP zeros compensate the dominant poles, the small signal model of superbuck converter resembles a conventional second-order system.

### 2.2. Model Variations to Deviated Parameters

The transfer function in (10) consistently changes with multiple parameters. Therefore, in practical operations, both zeros and poles might deviate from the original value. This would cause degraded transients and instability under conventional PI control.

As shown in Figure 4, the converter may suffer multiple parameter deviations, including circuit components of $\{L_1, L_2, C_1, C_2, R_d\}$ and input/output conditions of $\{R, v_{in}, v_{out}\}$. With deviated poles and zeros, the system may become unstable under conventional PI control, which can cause degraded transients and instability.

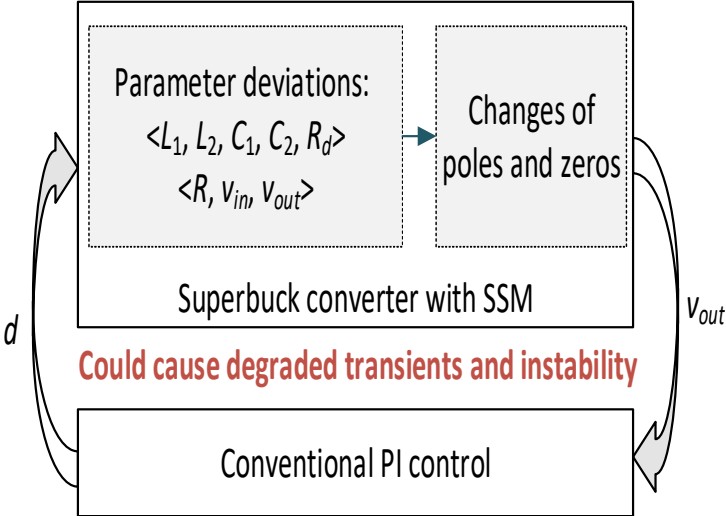

**Figure 4.** The superbuck converter under conventional PI control.

To address this issue, the controller should be dynamically adjusted, depending on the real-time parameters and operation conditions. However, directly calculating the required controller is relatively complex for the online implementation, and the reasons include:

- Owing to the complicated nonlinear fourth-order model, it is difficult to calculate the required controller on-line.
- Acquiring all the changing parameters online would induce complex hardware and sensing circuits.

Furthermore, a practical strategy to address the stability issue is to monitor the real-time stability based on the recent transients, and adaptively adjust the phase margin and bandwidth.

### 3. Transient Event-Capturing Circuit and Adaptive PI Control for the Superbuck Converter

The proposed controller is composed of a transient coupling circuit, an event-capturing circuit and an adaptive PI control. The transient coupling circuit and event-capturing circuit are connected into the feedback network, as shown in Figure 5.

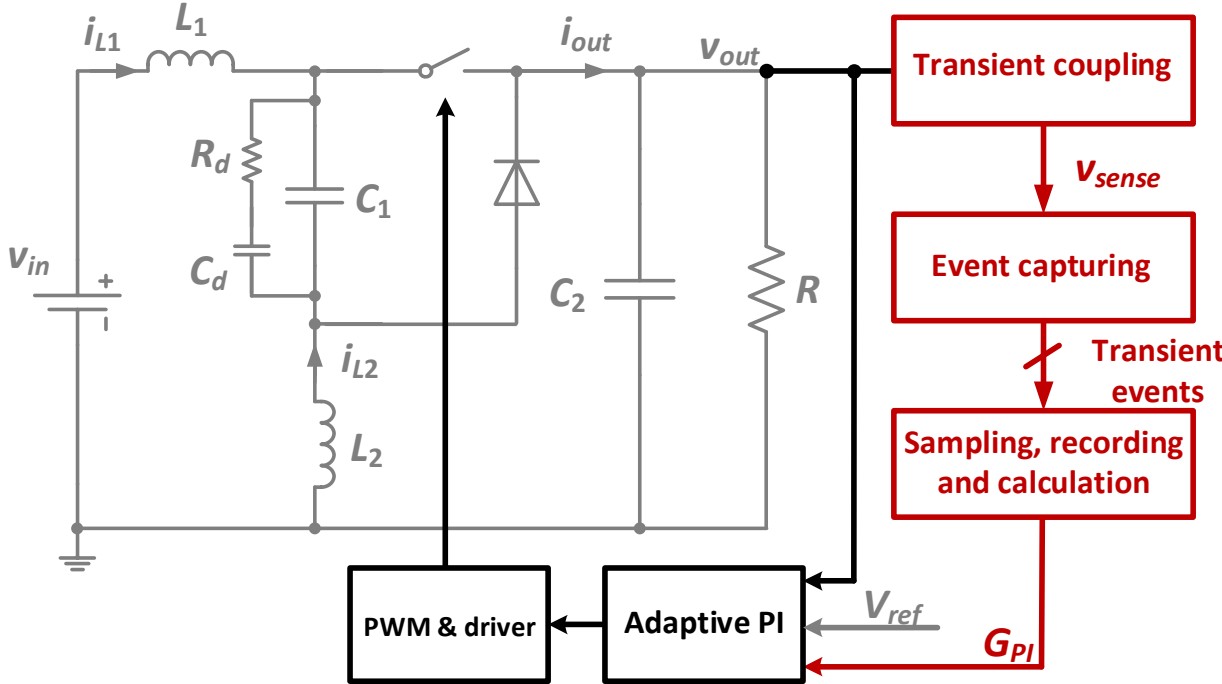

**Figure 5.** The superbuck converter with a transient coupling, an event-capturing and an adaptive PI control.

The transient coupling circuit converts the output voltage to the sensed voltage $v_{sense}$, which eliminates the DC component of output voltage. The event-capturing circuit detects up-crossings and down-crossings of $v_{sense}$, which indicate transient events of $v_{out}$. The captured events are sampled, recorded and processed to calculate the gain factor $G_{PI}$. Finally, $G_{PI}$ is sent to adaptive PI controller to adjust the phase margin and bandwidth.

To optimize the design, potential components in the output voltage are analyzed in the followings.

#### 3.1. Components in the Output Voltage

Output voltage of the superbuck converter is composed of DC, ripple and transient components. The DC component has been derived in (1). The ripple component is caused by the switching action, which is intrinsic in switched mode power supply converters. The transient component is the output voltage response for disturbances.

### 3.1.1. Transient Components

The transient component is analyzed by the loop gain, which depends on transfer functions of the superbuck power stage and the compensator. Under the condition of typical parameters and appropriate damping network, $G_{vd}(s)$ in (10) has four poles and two LHP zeros. When the LHP zeros compensate two dominant poles, the converter resembles a second-order system. As a result, the system can be well controlled with conventional PI compensation. The bode diagram is given by Figure 6a. To improve the transient response, the bandwidth should be slightly lower than nondominant poles, so that the phase margin is higher than 45 degrees.

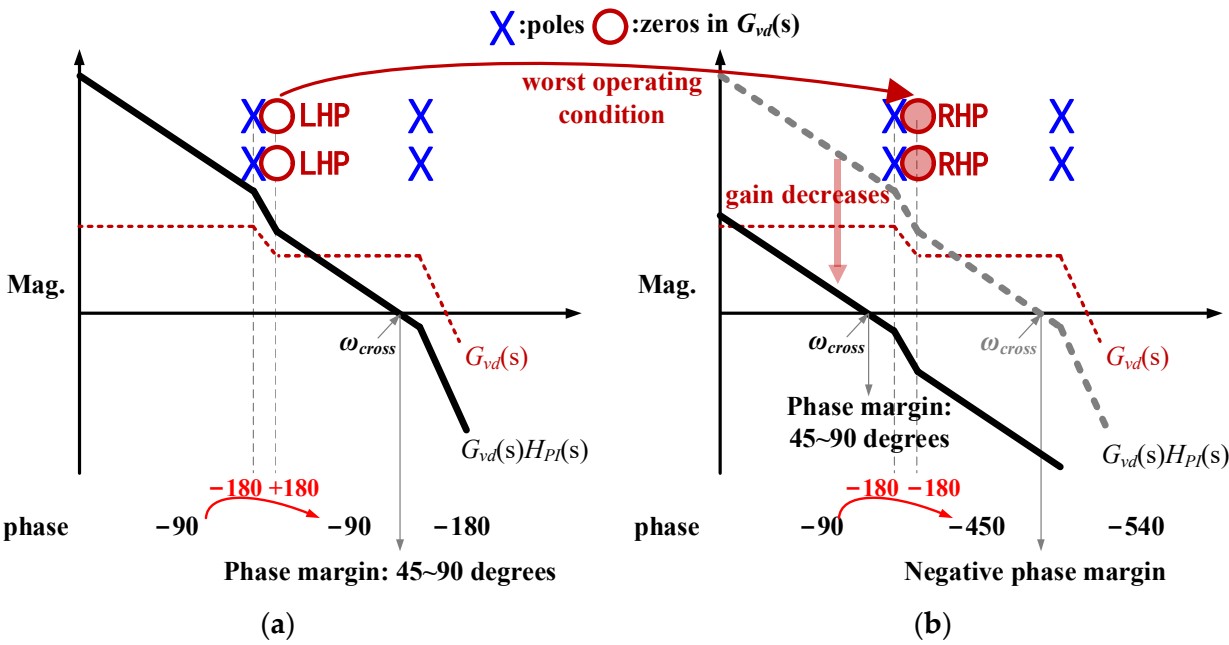

**Figure 6.** Bode diagrams of $G_{vd}(s)$ and $G_{vd}(s)H_{PI}(s)$ with (**a**) LHP zeros and (**b**) RHP zeros.

For the worst operating condition in Figure 6b, the LHP zeros in $G_{vd}(s)$ are moved to RHP. Two dominant poles change the phase by $-180$ degrees, while two RHP zeros further change the phase by $-180$ degrees. The phase margin can be negative that causes instability and oscillations in transients.

In order to improve the stability, the PI compensator needs to reduce the gain to decrease the bandwidth. The system can recover the stability when $\omega_{cross}$ is moved to lower than the dominant poles.

### 3.1.2. Ripple Components

The ripple components dependent on the output current, which is controlled by the duty ratio. Therefore, spectrum analysis is carried out with different duty ratios. To simplify the analysis, detailed calculations are provided with $D \approx 0$, $D \approx 1$ and $D = 0.5$.

For $D \approx 1$ and $D \approx 0$, spectrum of the output current is given in (11), where $I_{pp}$ is the inductance current peak-to-peak value.

$$\begin{cases} I_{out}(k\omega_{sw}) = \frac{1}{T}\int_{-T/2}^{T/2} \pm \frac{I_{pp}}{T} t e^{-jk\omega_{sw}t} dt = \pm \frac{I_{pp}}{T}\frac{(-1)^k}{jk\omega_{sw}} \\ |I_{out}(k\omega_{sw})| = \frac{I_{pp}}{T}\frac{1}{k\omega_{sw}} \end{cases}. \tag{11}$$

The magnitude is reciprocal to $k$, and it changes by $-20$ dB/dec with $k\omega_{sw}$. Furthermore, the output $RC$ network presents the low-pass characteristic, where the pole $1/(RC)$ is

usually much lower than $\omega_{sw}$. Therefore, transfer function of the network is approximate to $1/(sC)$, and magnitude of the output voltage is given by:

$$|V_{out}(k\omega_{sw})| \approx \left| \frac{I_{out}(k\omega_{sw})}{j\omega C} \right| = \frac{I_{pp}}{CT} \frac{1}{k^2\omega_{sw}^2}. \tag{12}$$

For $D = 0.5$, spectrum of $i_{out}$ is given by (13):

$$I_{out}(k\omega_{sw}) = \frac{1}{T}\int_0^{T/2}\left(\frac{2I_{pp}}{T}t - \frac{I_{pp}}{2}\right)e^{-jk\omega_{sw}t}dt + \frac{1}{T}\int_{-T/2}^0\left(-\frac{2I_{pp}}{T}t - \frac{I_{pp}}{2}\right)e^{-jk\omega_{sw}t}dt = \frac{4I_{pp}}{T^2}\frac{(-1)^k - 1}{k^2\omega_{sw}^2}. \tag{13}$$

The magnitude is zero when $k$ is even, and it is reciprocal to $k^2$ when $k$ is odd. Furthermore, by the *RC* network, the magnitude of the output voltage is given by:

$$|V_{out}(k\omega_{sw})| \approx \frac{4I_{pp}}{CT^2}\frac{(-1)^k - 1}{k^3\omega_{sw}^3}. \tag{14}$$

According to (12) and (14), magnitudes of the output voltage are plotted in Figure 7.

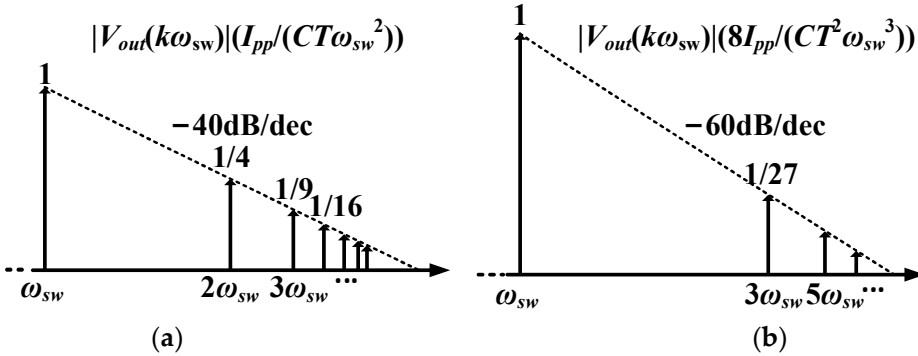

**(a)**                 **(b)**

**Figure 7.** Magnitudes of the output voltage (**a**) when $D \approx 0$ or $D \approx 1$, and (**b**) when $D = 0.5$.

As $k\omega_{sw}$ increases, $|V_{out}(k\omega_{sw})|$ changes by $-40$ dB/dec when $D \approx 1$ and $D \approx 0$, and changes by $-60$ dB/dec when $D = 0.5$. The ripple components are mainly composed of the fundamental wave, the second harmonic wave and the third harmonic wave. For higher than fourth harmonics, the magnitudes are consistently lower than $1/16$ of the fundamental wave.

### 3.2. The Transient Coupling Circuit

The transient coupling circuit aims to acquire the transient signals. To eliminate the DC and ripple components, a band-pass filter is required and shown in Figure 8a. The transient coupling circuit has an impedance $Z_1$, which blocks signals outside the passband. Range of the passband is dominated by the relationship between $|Z_1|$ and the main sensing resistor $R_s$. The transient coupling circuit forms a low-impedance pass to the ground, which is used to eliminate the ripple component at the switching frequency.

To achieve the fix gain within the passband, it is suggested to set $R_r = R_s$. Further, to reduce the influence of coupling circuit on the power stage, the impedance within the passband should be much higher than the load resistance, i.e., $R_r + R_s \gg R$. Complete transfer function of the transient coupling circuit is complicated. However, it can be simplified at different frequencies.

At the low frequency range, since $Z_2$ is dominated by $R_s$, the transfer function is approximate to

$$H_{sense}(s)|_L \approx \frac{sR_sC_s}{s(R_s + R_r)C_s + 1}. \tag{15}$$

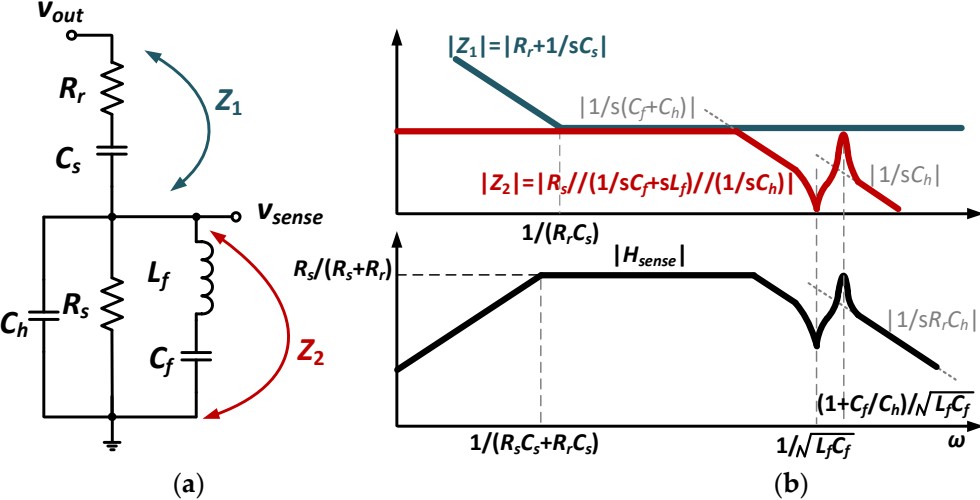

**Figure 8.** The transient coupling circuit: (**a**) the circuit scheme and (**b**) the magnitude–frequency characteristic.

As the frequency increases, $C_s$ is equivalent to the short circuit. For the frequency lower than $1/\sqrt{L_f C_f}$, $Z_2$ is approximate to $R_s/(1 + sR_s(C_f + C_h))$. Therefore, $H_{sense}(s)$ is approximate to

$$H_{sense}(s)|_M \approx \frac{R_s}{s(C_f + C_h)R_r R_s + R_s + R_r}. \tag{16}$$

Based on (15) and (16), low and high frequencies of the passband are given by

$$\begin{cases} \omega_L = \frac{1}{(R_s + R_r)C_s} \\ \omega_H = \frac{R_s + R_r}{(C_f + C_h)R_r R_s} \end{cases}. \tag{17}$$

For the frequency higher than $\omega_H$, the low impedance path in $Z_2$ is formed by $L_f$-$C_f$ or $C_h$. Neglecting $R_s$, $H_{sense}(s)$ is approximate to

$$H_{sense}(s)|_H \approx \frac{1}{sR_r C_h} \frac{s^2 L_f C_f + 1}{s^2 L_f C_f + 1 + C_f/C_h}. \tag{18}$$

(18) has conjugate poles at $\sqrt{(1 + C_f/C_h)/(L_f C_f)}$, and conjugate zeros at $\sqrt{1/(L_f C_f)}$. The conjugate zeros should be at the switching frequency to eliminate the fundamental wave. The conjugate poles are higher than the conjugate zeros, and they should not be at the harmonic frequencies. Otherwise, the poles can enhance unfavorable harmonics. The amplitude between the conjugate poles and the conjugate zeros is $\sqrt{(C_h + C_f)/C_h}$, which should be as close to 1 as possible to avoid enhancing the harmonics. Further, since (18) is approximate to $1/(sR_r C_h)$ when the frequency is higher than the conjugate poles, a high $C_h$ can provide a better reduction for the harmonics.

Finally, the conjugate poles in (18) are 1.2 fold the conjugate zeros. The above design rules are summarized as:

$$\begin{cases} R_r = R_s, \ R_s + R_r >> R \\ C_s = \frac{1}{(R_s + R_r)\omega_L}, \ C_h = \frac{1}{1.44} \frac{R_s + R_r}{\omega_H R_r R_s} \\ L_f = \frac{1.44}{0.44} \frac{\omega_H}{\omega_{sw}^2} \frac{R_r R_s}{R_s + R_r}, \ C_f = \frac{0.44}{1.44} \frac{R_s + R_r}{\omega_H R_r R_s} \end{cases}. \tag{19}$$

Under the condition of (19), the gain within passband is $R_s/(R_s + R_r)$. The gain at switching frequency is infinitely small, while gains at the $k^{th}$ ($k \geq 2$) harmonic are $1/(k\omega_{sw}R_r C_h)$.

### 3.3. The Event-Capturing Circuit

The event-capturing circuit aims to capture and classify transient events in $v_{sense}$. As shown in Figure 9a, it is composed of two complementary branches, which are symmetric to detect up-crossings and down-crossings of $v_{sense}$, respectively. Each branch is composed of a relay unit, a rising edge detector, a *RC* circuit and a resettable counter. The final logic unit processes the counter outputs, and outputs single-bit event signals.

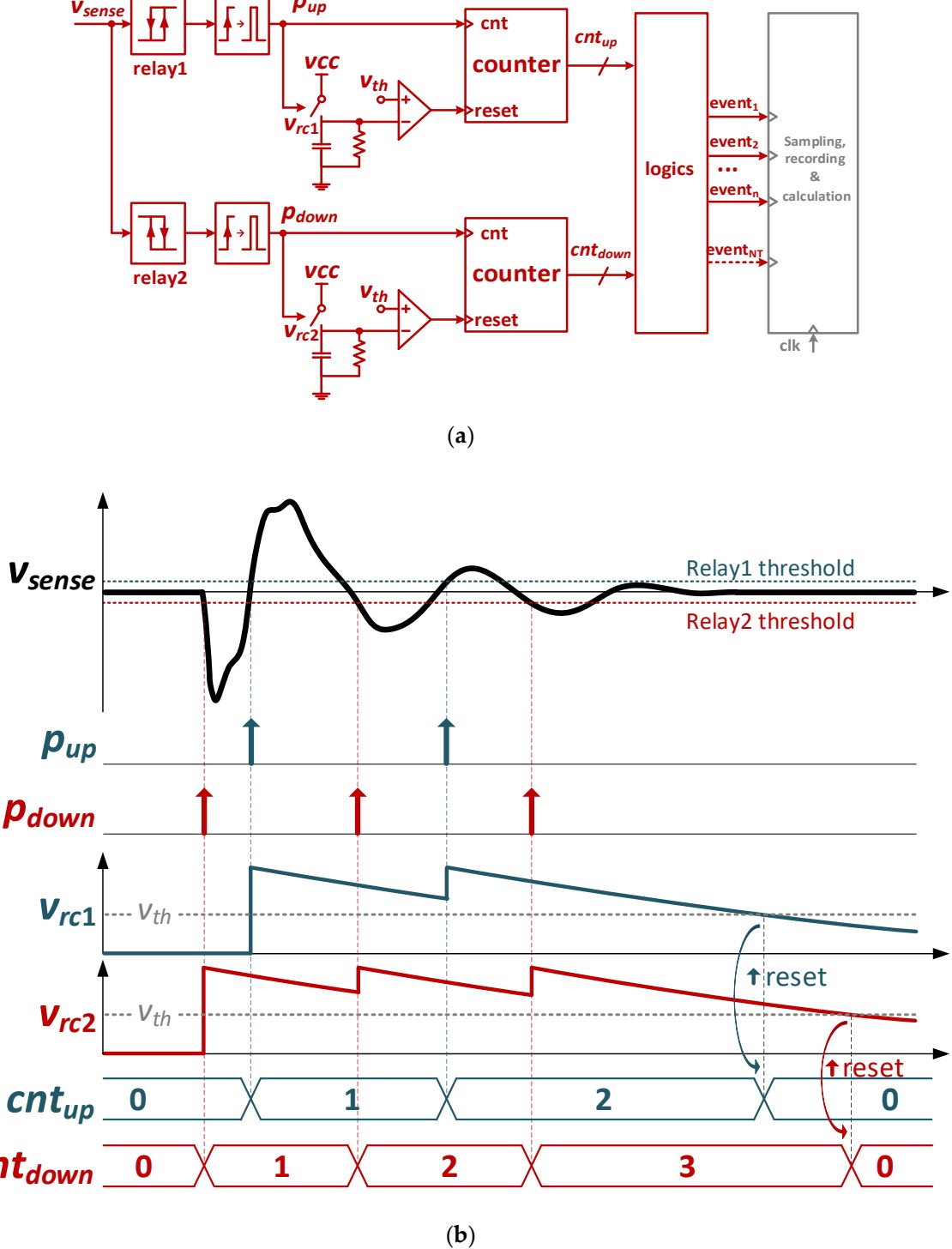

**Figure 9.** The event-capturing circuit: (**a**) the circuit scheme and (**b**) the capturing process.

A typical capturing process is shown in Figure 9b. When $v_{sense}$ crosses the relay threshold, the branch detects the crossing and generates a narrow pulse, i.e., $p_{up}$ and $p_{down}$. The following counters count the up-crossing and down-crossing times by narrow pulses, denoted as $cnt_{up}$ and $cnt_{down}$. The counters are reset when the *RC* circuits generate rising pulses at the *RC* voltage below $v_{th}$. Finally, the DSP captures and classifies transient events based on the counter outputs.

Bit width of the counters determines event kinds in the capturing process. Bit width should be designed with consideration of the power consumption, the circuit cost and the memory size, etc. With the 2 bits capturing circuit, potential transient events are illustrated in Figure 10.

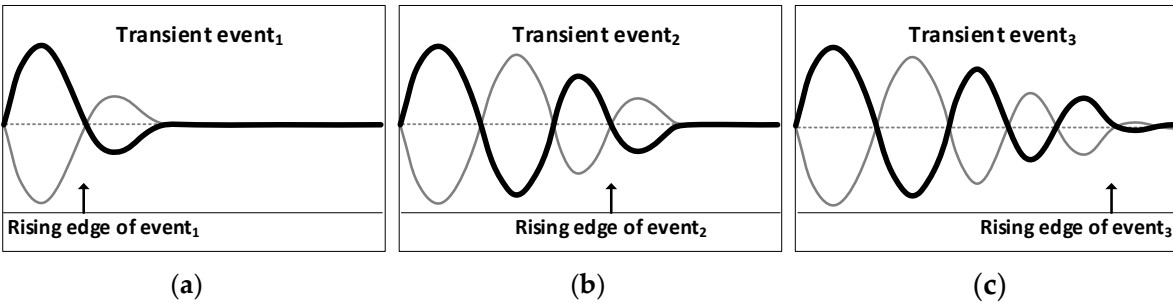

**Figure 10.** The potential transient events with the 2 bits capturing circuit: (**a**) event$_1$ oscillates for 1~1.5 cycles, (**b**) event$_2$ oscillates for 2~2.5 cycles, and (**c**) event$_3$ oscillates for more than 3 cycles.

When $cnt_{up} = 2'b01$ and $cnt_{down} = 2'b01$, the event$_1$ signal is captured, which oscillates for 1~1.5 cycles. When $cnt_{up} = 2'b10$ and $cnt_{down} = 2'b10$, the event$_2$ is captured, which oscillates for 2~2.5 cycles. When $cnt_{up} = 2'b11$ and $cnt_{down} = 2'b11$, the event$_3$ is captured, which lasts for more than three cycles. More fine-grained events can be captured and classified by increasing the bit width. To transform $cnt_{up}$ and $cnt_{down}$ as event signals, the required logic table is given in Table 1.

**Table 1.** Logics of the event-capturing circuit.

| $Cnt_{up}$ | $Cnt_{down}$ | $Event_1$ | $Event_2$ | $Event_3$ |
|:---:|:---:|:---:|:---:|:---:|
| 00 | 00 | 0 | 0 | 0 |
| 00 | 01 | 0 | 0 | 0 |
| 00 | 10 | 0 | 0 | 0 |
| 00 | 11 | 0 | 0 | 0 |
| 01 | 00 | 0 | 0 | 0 |
| 01 | 01 | 1 | 0 | 0 |
| 01 | 10 | 0 | 0 | 0 |
| 01 | 11 | 0 | 0 | 0 |
| 10 | 00 | 0 | 0 | 0 |
| 10 | 01 | 0 | 0 | 0 |
| 10 | 10 | 0 | 1 | 0 |
| 10 | 11 | 0 | 0 | 0 |
| 11 | 00 | 0 | 0 | 0 |
| 11 | 01 | 0 | 0 | 0 |
| 11 | 10 | 0 | 0 | 0 |
| 11 | 11 | 0 | 0 | 1 |

Based on Table 1, the logic outputs are derived as

$$\begin{cases} event_1 = \overline{cnt_{up}[1]} \bullet cnt_{up}[0] \bullet \overline{cnt_{down}[1]} \bullet cnt_{down}[0] \\ event_2 = cnt_{up}[1] \bullet \overline{cnt_{up}[0]} \bullet cnt_{down}[1] \bullet \overline{cnt_{down}[0]} \\ event_3 = cnt_{up}[1] \bullet cnt_{up}[0] \bullet cnt_{down}[1] \bullet cnt_{down}[0] \end{cases} . \tag{20}$$

### 3.4. The Adaptive PI Control Based on the Captured Events

Based on the transient event capturing, an adaptive PI control is designed to improve the stability with deviated parameters, as shown in Figure 11. The controller utilizes $G_{PI}$ to adjust the loop gain, where $G_{PI}$ is calculated with weightings of each captured event.

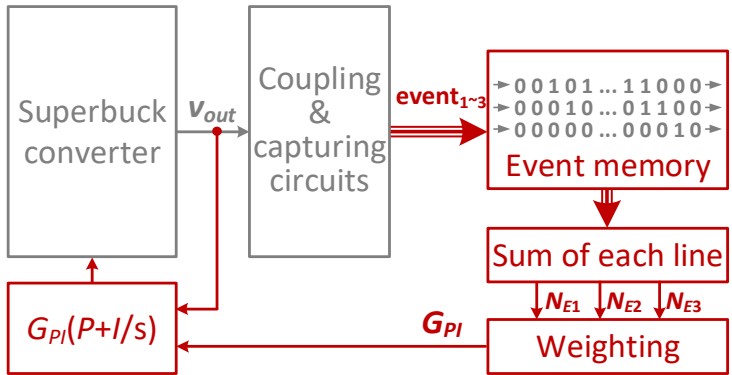

**Figure 11.** An adaptive PI control based on transient event capturing.

After transient coupling and event capturing, events are buffered and sampled into the event memory. The memory follows first-in-first-out (FIFO) principle, and continuously accepts new data. Each line of event memory indicates a kind of event in a recent period of time (namely recording window). After that, the controller sums each line to calculate the numbers of each kind of event, i.e., $\{N_{E1}, N_{E2}, N_{E3}\}$. Furthermore, with the consideration of weighting factors, the final gain for the adaptive PI control is calculated as:

$$G_{PI} = \frac{1}{1 + k_1 N_{E1} + k_2 N_{E2} + k_3 N_{E3}}. \tag{21}$$

where $\{k_1, k_2, k_3\}$ are weighting factors, and $G_{PI} \leq 1$. The higher weighting factor represents the higher sensitive for the captured event. The controller achieves the highest bandwidth when $\{N_{E1}, N_{E2}, N_{E3}\}$ are all zeros. When transient events occur, the reduced $G_{PI}$ can reduce the bandwidth and improve the phase margin. This is benefit for improving the stability in transients.

The weighting factors can be derived with the following procedures:

- Design the recording window of event memory. This determines how long the recent events can affect the adaptive control. The length affects the maximum number of captured events.
- Design the reduced $G_{PI}$ as expected in the recording window. This should be set with tradeoff between response speed and stability. For example, assuming 30 event$_1$ or 20 event$_2$ in the window can reduce $G_{PI}$ by half, then $k_1$ and $k_2$ should be set as $1/30$ and $1/20$.

## 4. Simulations

### 4.1. The Root Locus of $G_{vd}(s)$ with Deviated Parameters

The followings provide the simulated root locus of $G_{vd}(s)$ with deviated parameters. Specifications under the typical operation are given in Table 2.

**Table 2.** Specifications of the superbuck converter.

| | | | |
|---|---|---|---|
| $L_1$ | 250 µH | $P$ | 0~100 W |
| $L_2$ | 110 µH | $P_{CPL}$ | 0~50 W |
| $C_1$ | 2.5 µF | $v_{in}$ | 42 V |
| $C_2$ | 7 µF | $v_{out}$ | 28 V |
| $f_{sw}$ | 100k Hz | | |

Figure 12a provides the root locus when $R_d$ changes from 8 Ω to 50 Ω. Two dominant poles are conjugated from −22 Krad/s to −8 Krad/s, while two nondominant poles are conjugated from −18 Krad/s to −12 Krad/s. Two zeros are conjugated from −18 Krad/s to 6 Krad/s. As all poles and zeros increases, the damping factors decrease, which indicates the decreased stability. When $R_d$ achieves a critical value, the conjugated zeros move to the RHP that fail to compensate the poles.

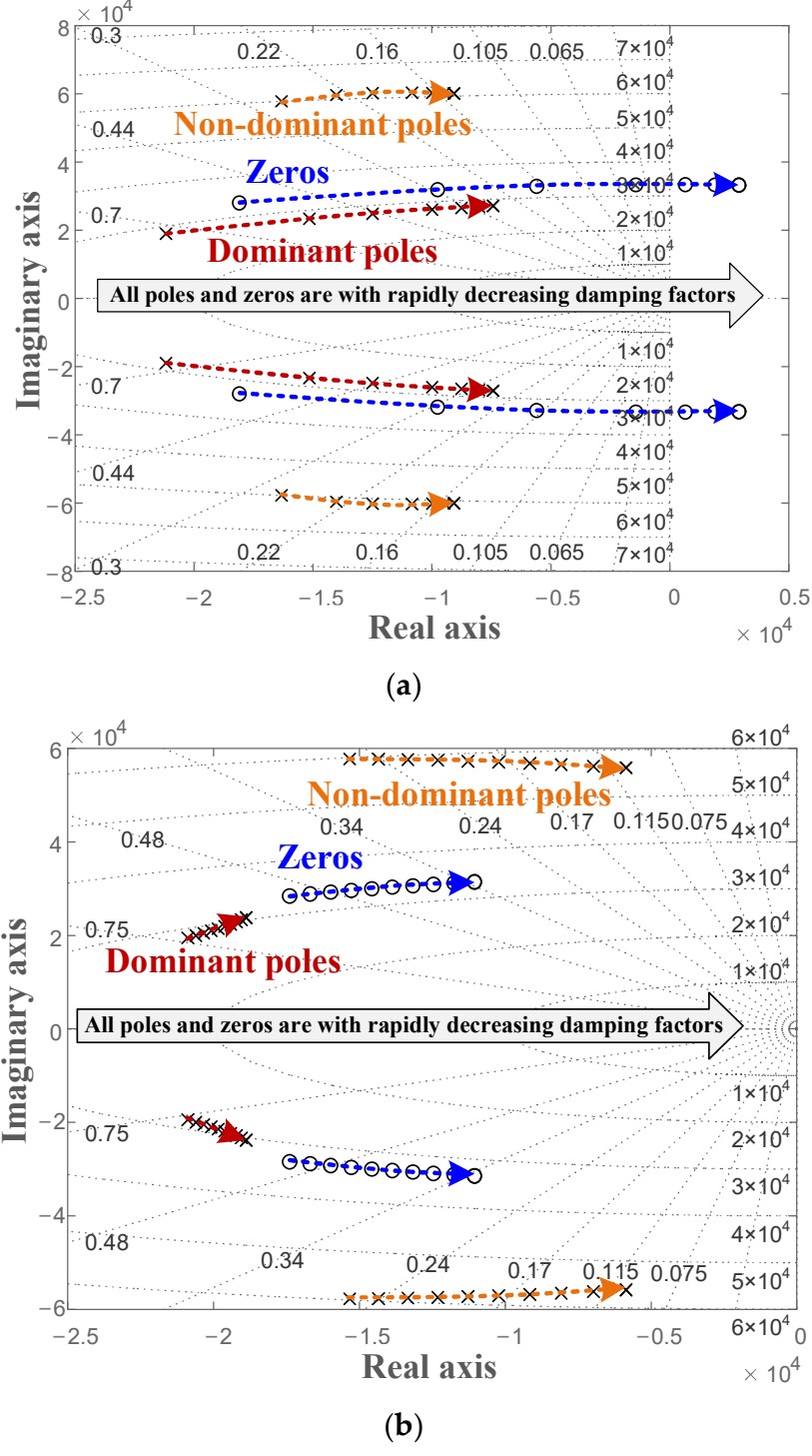

**Figure 12.** Root locus of $G_{vd}(s)$ (**a**) when $R_d$ changes from 8 Ω to 50 Ω; (**b**) when $P_{CPL}$ changes from 10 W to 100 W.

Figure 12b provides the root locus when $P_{CPL}$ changes from 10 W to 100 W (the resistive load is 50 W). On one hand, the conjugated zeros move apart from the dominant poles. On the other hand, as all poles and zeros increases, the damping factors decrease. This decreases the phase margin that harms the stability.

*4.2. The Frequency Response of the Coupling Circuit*

The frequency response of the coupling circuit is simulated to verify the design in Section 3.2. To pass transient signals, the passband is set as $\omega_L = 10$ k rad/s and $\omega_H = 210$ k rad/s. Furthermore, under design rules in (19), parameters of the coupling circuits are adjusted as $R_s = 200\ \Omega$, $C_s = 250$ *nF*, $L_f = 175\ \mu$H, $C_f = 14.6$ *nF*, $C_h = 33.2$ *nF* and $R_r = 200\ \Omega$. The impedance within the passband is 400 $\Omega$, which is much higher than the load resistance. This is beneficial to reduce the AC current flowing through the coupling circuit. Furthermore, the frequency response of the coupling circuit is plotted in Figure 13.

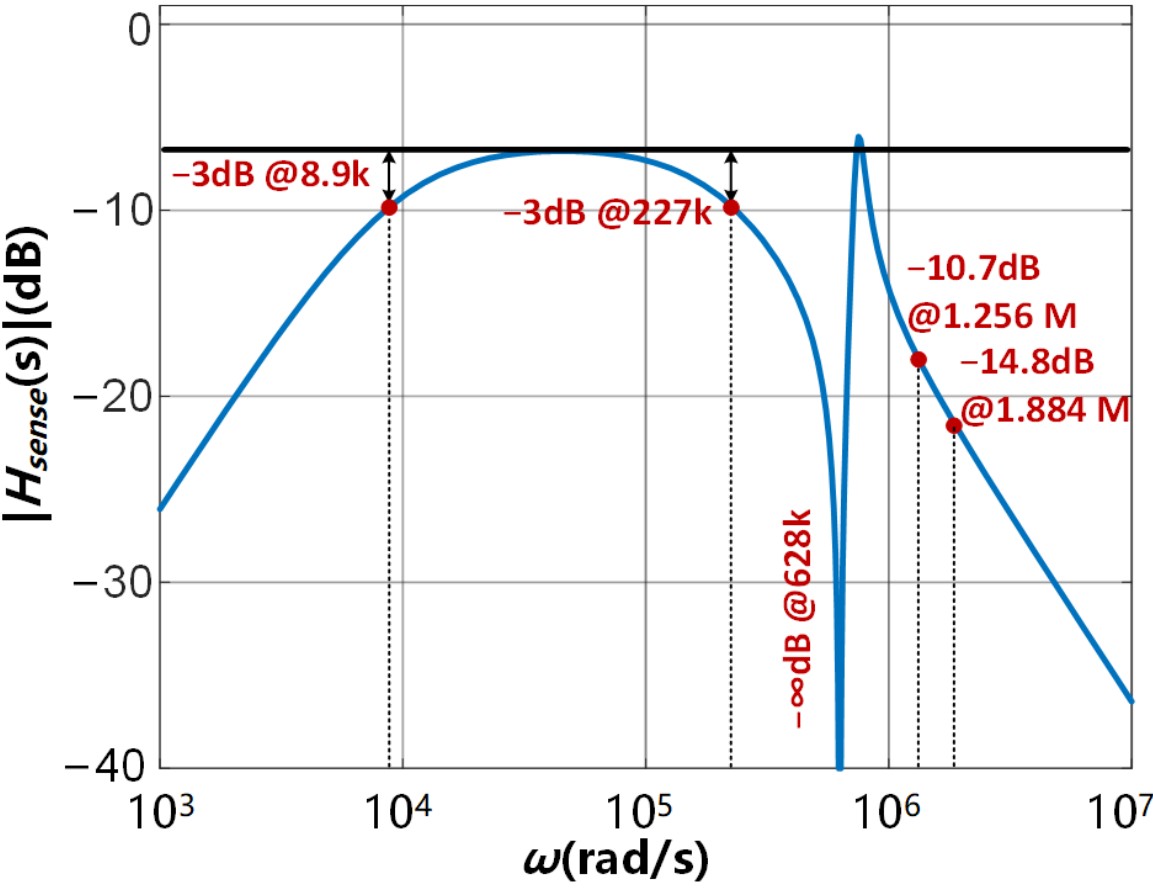

**Figure 13.** Frequency responses of the coupling circuit.

The achieved pass band is within 8.9 krad/s and 227 krad/s. It has infinite damping at the fundamental switching frequency, which is benefit for the ripple reduction. The damping factors to the second and third harmonics are −10.7 dB and −14.8 dB, respectively.

*4.3. The transient Comparing between an Adaptive PI Controller and an Adaptive Phase Margin*

The SIMULINK simulates the output voltage responses of an adaptive PI controller and an adaptive phase margin controller. The output voltage responses are shown in Figure 14 when the reference voltage changes from 25 V to 28 V, the line voltage changes from 37 V to 42 V and the load changes from 10 $\Omega$ to 8 $\Omega$. Comparing with the adaptive phase margin controller, response time and overshoot of the adaptive PI controller are smaller. The red lines in Figure 14a–c represent reference voltage, line voltage and load changings, respectively.

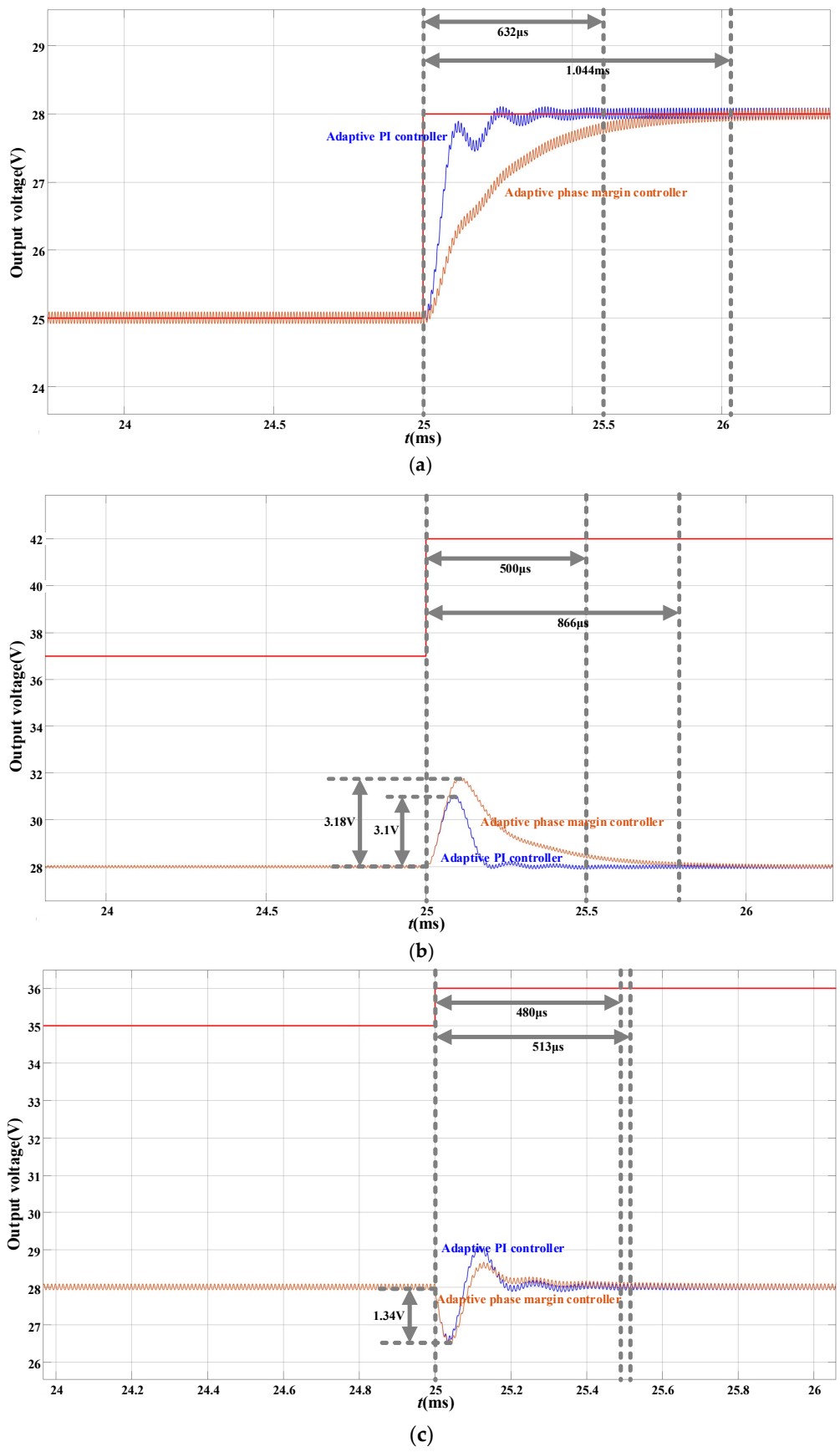

**Figure 14.** The output voltage responses when (**a**) the reference voltage changes from 25 V to 28 V, (**b**) the line voltage changes from 37 V to 42 V and (**c**) the load changes from 10 Ω to 8 Ω.

## 5. Experiments

An experimental prototype is carried out to verify effectiveness of the proposed strategy. The photograph is shown in Figure 15, and the main specifications are given in Table 2. The coupling and capturing circuits are realized on individual PCB boards with discrete components, which are TLV3502 (hysteresis comparator), SN74LV393A (counter), SN74AUP2G02 (monostable trigger) and ADG721 (switch). The logic circuit is realized with four SN74 chips (SN74LVC00A, SN74LV21A, SN74LV11A and SN74LV32A). PCB boards of the superbuck converter, digital logics and transient coupling and event-capturing circuits occupy areas of 7 cm × 9 cm, 3 cm × 3.1 cm and 3 cm × 2.8 cm, respectively.

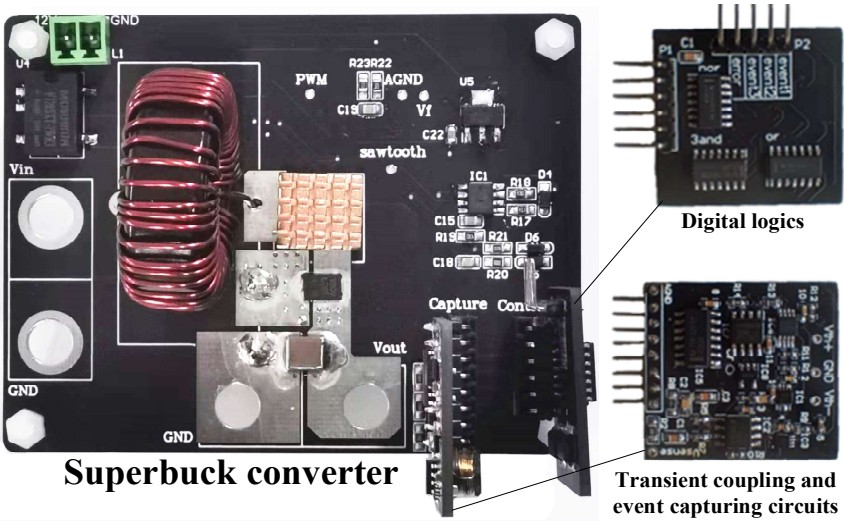

**Figure 15.** The experimental prototype.

In the followings, relevant results are measured with repetitive steps of the reference voltage. All events are buffered and sampled by a rate of 1 kHz, while the memory records all events in the recent 50 ms (recording window). The adaptive PI compensator is set as [$P = 0.006\ G_{PI}$, $I = 350\ G_{PI}$], and $G_{PI} = 1/(1 + 0.03 \times N_{E1} + 0.02 \times N_{E2} + 0.02 \times N_{E3})$. With the captured events, $G_{PI}$ is adjusted and clamped within [0.4, 1].

### 5.1. Transients with the Conventional PI Controller

At $G_{PI} = 1$, the adaptive PI controller is equivalent to the conventional PI with [$P = 0.006$, $I = 350$]. In this experiment, the controller achieves the maximum bandwidth. Relative results are shown in Figure 16. The coupling and capturing circuits are still working. The coupling circuit eliminates the DC component in $v_{out}$, and effectively reduces the ripple component. Pulses in $p_{up}$ and $p_{down}$ indicate up-crossings and down-crossings of $v_{sense}$, respectively.

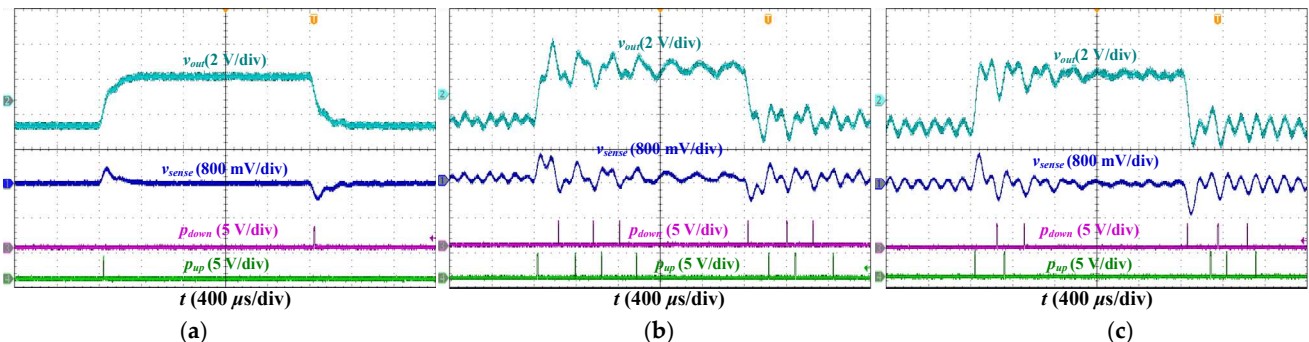

**Figure 16.** Transients at $G_{PI} = 1$ (**a**) under typical parameters, (**b**) when $R_d$ deviates to 50 Ω, and (**c**) when 50 W of the load is switched to CPL.

In Figure 16a, the result is measured under typical parameters. When $v_{ref}$ steps from 25 V to 28 V, the output voltage follows the reference voltage without oscillations. No event is captured. In Figure 16b, the result is measured under $R_d = 50\ \Omega$. The deviation induces RHP zeros and degrades the stability. As a result, event$_3$ is captured when $v_{ref}$ steps from 25 V to 28 V. In Figure 16c, the result is measured when half of the load is CPL (both resistive and CPL loads are 50 W). As a result, event$_2$ and event$_3$ are captured during the up-stepping and down-stepping, respectively.

*5.2. Transients with an Adaptive PI controller under Deviated $R_d$*

When using an adaptive PI control, $G_{PI}$ is dynamically adjusted according to the captured events. Furthermore, when $R_d$ deviates to 50 $\Omega$, the measured results are given in Figure 17.

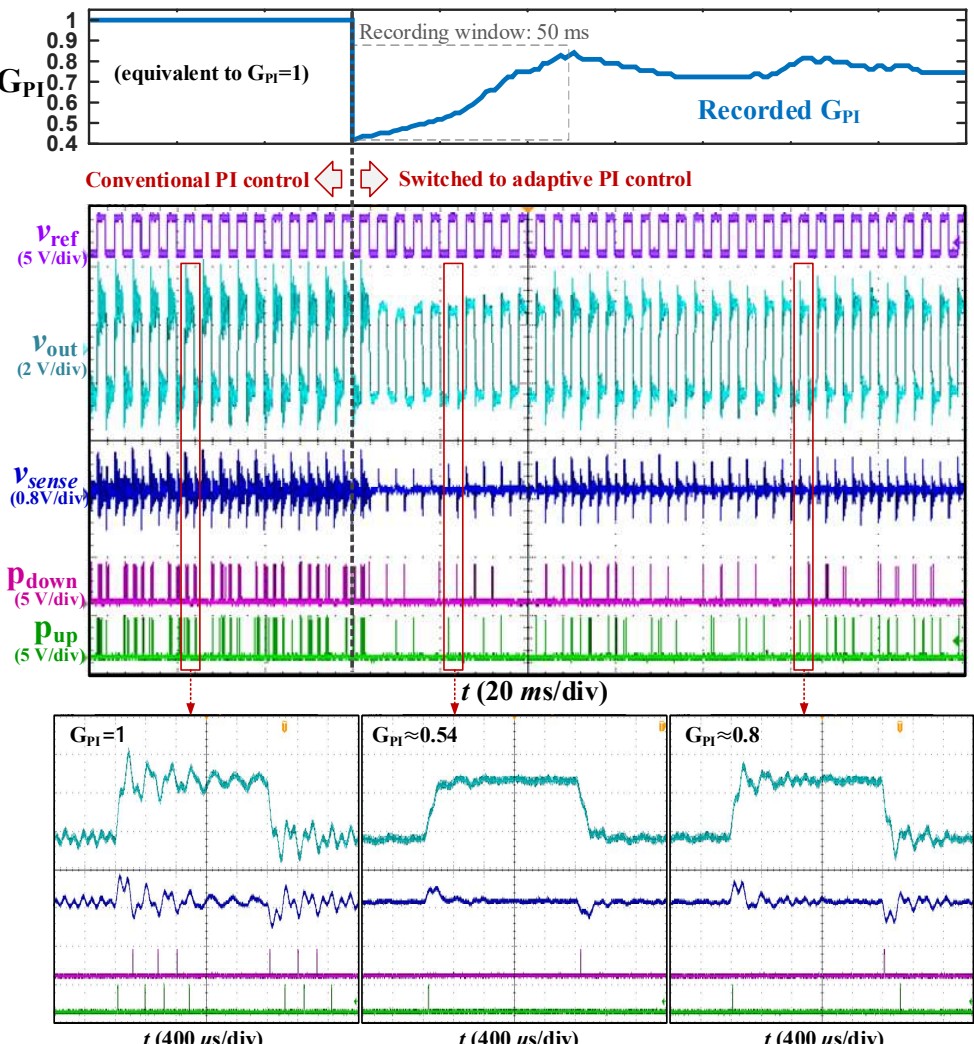

**Figure 17.** Transients with an adaptive PI controller when $R_d$ is deviated to 50 $\Omega$.

Before switching to the adaptive PI control, the output voltage dramatically oscillates in the transients, leading to multiple pulses in $p_{up}$ and $p_{down}$. The captured events lead to a small $G_{PI}$. When the adaptive PI control is switched on, the transient response is immediately improved. Oscillations are almost eliminated, which greatly reduces the pulses in $p_{up}$ and $p_{down}$. However, as the captured events in the recording window decrease, $G_{PI}$ gradually increases and finally stabilizes within [0.7, 0.8]. The decreased $G_{PI}$ effectively improves the transient response and stability.

### 5.3. Transients with an Adaptive PI Controller under Partial Constant Power Load

When half of the load is CPL (both resistive and CPL loads are 50 W), the measured results are given by Figure 18.

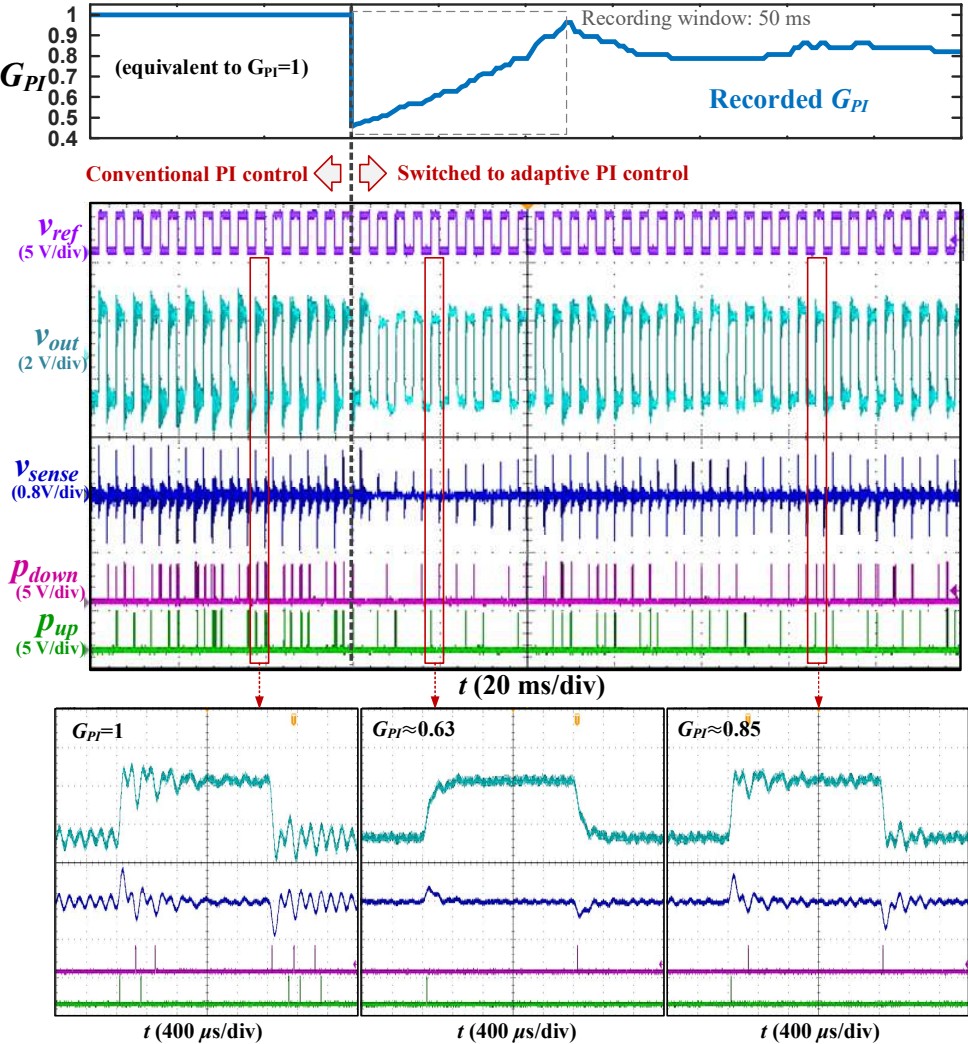

**Figure 18.** Transients with an adaptive PI controller when 50 W of the load is switched to CPL.

The $G_{PI}$ adjustment in Figure 18 has the similar process as that in Figure 17. Before the adaptive PI is switched on, multiple transient oscillations occur that lead to a small $G_{PI}$. When the adaptive PI control is switched on, the transient response is improved by a low $G_{PI}$. As the captured events in the recording window decrease, $G_{PI}$ gradually increases and finally stabilizes around 0.8. All results prove effectiveness of the proposed strategy for improving the transient response with deviated parameters.

### 5.4. Transients with an Adaptive PI Controller at Load and Line Steps

When the load changes from 8 Ω to 10 Ω and from 10 Ω to 8 Ω ($R_d = 8$ Ω, $v_{in} = 42$ V, $v_{out} = 28$ V), the measured results are shown in Figure 19a,b. The response time and overshoots are 310 μs and 440 mV at load changing from 8 Ω to 10 Ω, and 300 μs and 420 mV at load changing from 8 Ω to 10 Ω, respectively. Before switching to the adaptive PI control, the output voltage dramatically oscillates in the transients, leading to 1 up-crossing and 1 down-crossing. Then, the event$_1$ leads to the adaptive PI control. The transient response is immediately improved, and oscillations are almost eliminated.

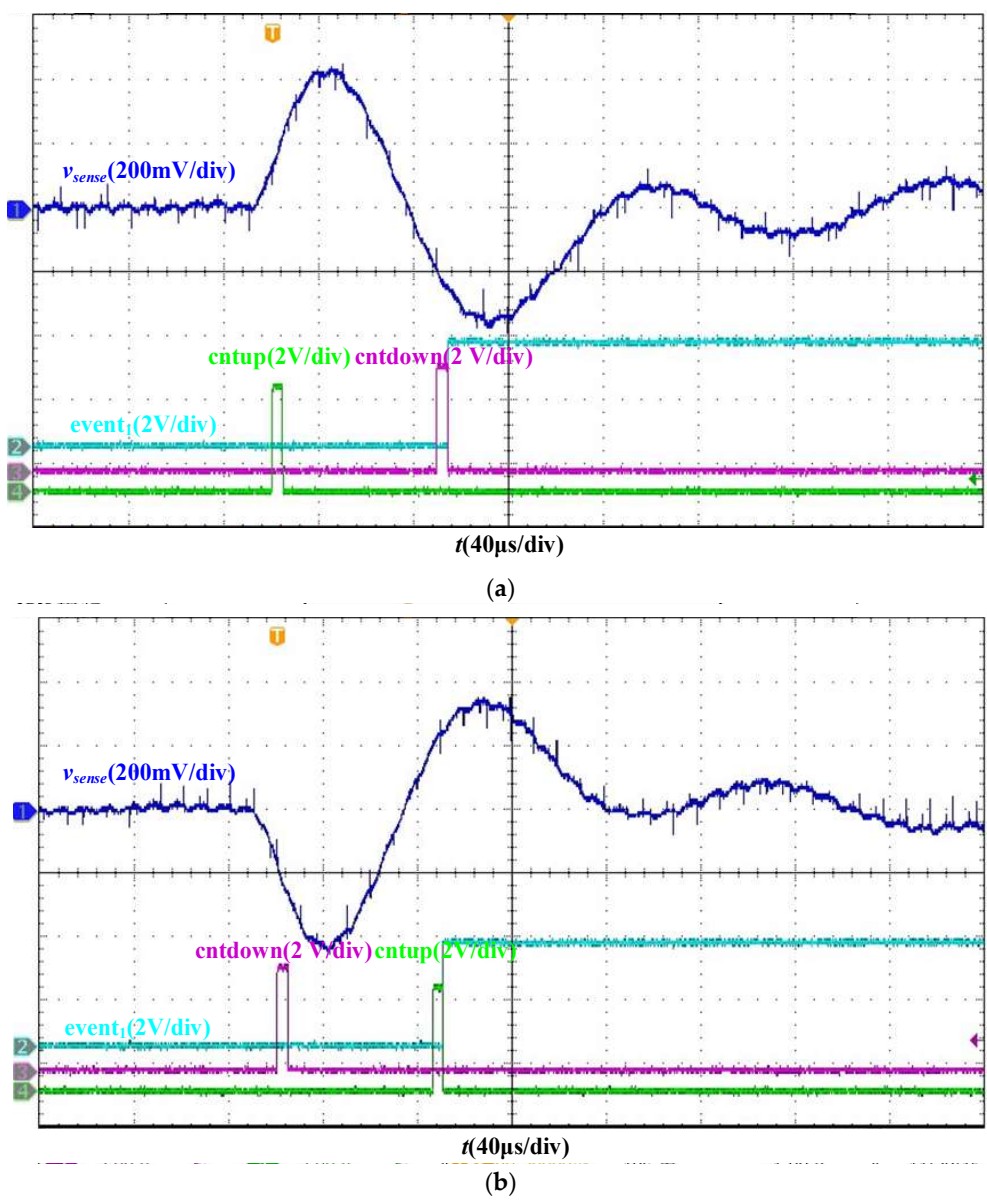

**Figure 19.** Transients for the load changing (**a**) from 8 Ω to 10 Ω and (**b**) from 10 Ω to 8 Ω.

When the line voltage changes from 42 V to 37 V ($R_d$ = 8 Ω, $R$ = 10 Ω, $v_{out}$ = 28 V), the measured results are shown in Figure 20. The response time and downshoot are 770 μs and 430 mV. Before switching to the adaptive PI control, the output voltage dramatically oscillates in the transients, leading to 1 up-crossing and 1 down-crossing. Then, the $event_1$ leads to the adaptive PI control, the transient response is immediately improved, and oscillations are almost eliminated.

### 5.5. Power Loss Estimation

Power loss of the superbuck converter is mainly composed of a switch, a diode and two inductors. To estimate the losses, the power losses of a switch, a diode and two inductors are calculated at 100 W and 50 W. The power losses of a switch, a diode and two inductors are given by

$$P_{switch} = P_{ds} + P_{on/off} = I_o{}^2 R_{ds} D + \frac{f_{sw}(E_{on} + E_{off})i_o}{20} + \frac{1}{2}\frac{f_{sw}(v_{in} + v_o)E_{oss}}{50}$$
$$P_{diode} = P_f = V_F I_o (1 - D)$$
$$P_{inductor1} = P_{coil1} + P_{core1} = I_{L1}{}^2 \rho_{Cu}\frac{l_1}{S} + 4.5415 B_m{}^{2.1236} f_{sw} + 0.0227(B_m f_{sw})^2$$
$$P_{inductor2} = P_{coil2} + P_{core2} = I_{L2}{}^2 \rho_{Cu}\frac{l_2}{S} + 4.5415 B_m{}^{2.1236} f_{sw} + 0.0227(B_m f_{sw})^2$$

$$(22)$$

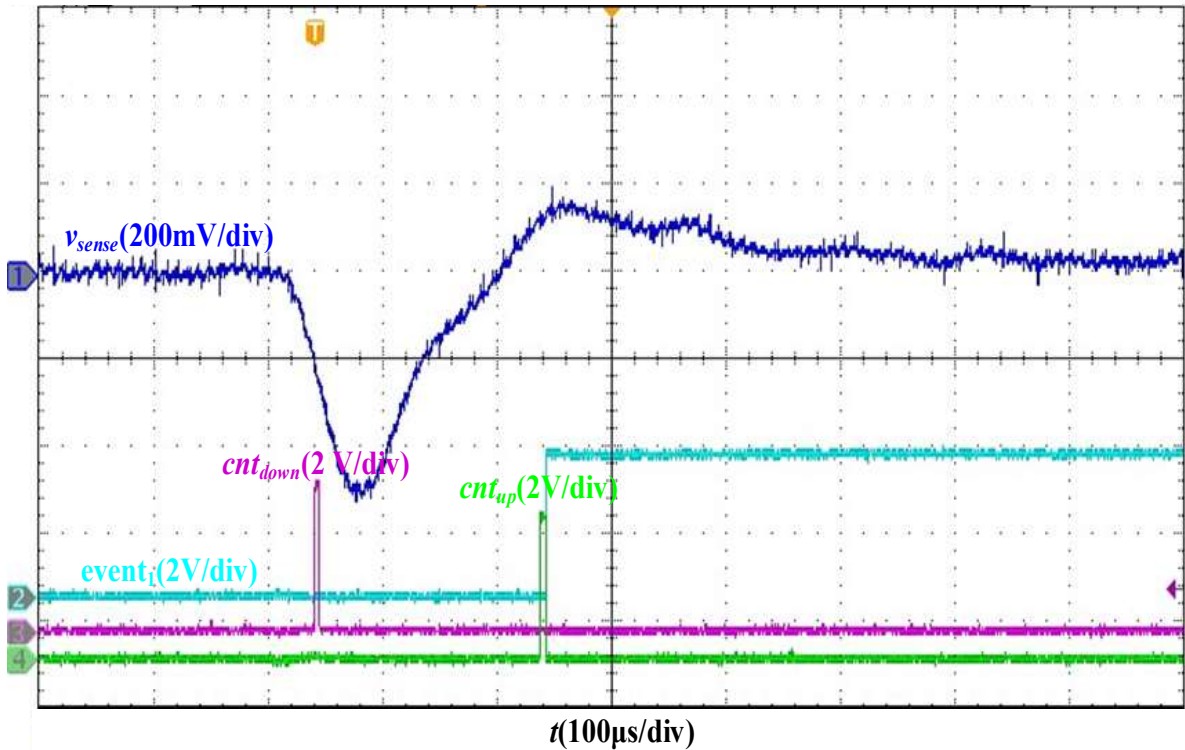

**Figure 20.** Transients for the line voltage changing from 42 V to 37 V.

Therefore, power loss estimation at 100 W and 50 W are shown in Table 3.

**Table 3.** Power loss estimation.

| Power Level (W) | $P_{loss,total}$ (W) | $P_{switch}$ (W) | $P_{diode}$ (W) | $P_{inductor1}$ (W) | $P_{inductor2}$ (W) | Efficiency |
|---|---|---|---|---|---|---|
| 100 | 1.515 | 0.251 | 0.863 | 0.223 | 0.178 | 98.5% |
| 50 | 0.908 | 0.166 | 0.432 | 0.167 | 0.143 | 98.2% |

## 6. Conclusions

This paper presents a transient coupling circuit and an event-capturing circuit for the voltage mode superbuck converter, along with an adaptive PI control. The transient coupling circuit converts the output voltage to the sensed voltage, which eliminates the DC and ripple components in the output voltage. By counting up-crossing and down-crossing times of the sensed voltage, the capturing circuit captures and classifies different events that indicate the real-time stability. The proposed coupling and capturing circuit is simple to realize and cost effective, which are implemented by the passive components and logic chips. Furthermore, based on the transient event capturing, an adaptive PI controller is designed to adjust the loop gain. Experimental results of a 100 W superbuck converter verify effectiveness of the adaptive PI controller for improving the transient response and stability. Adaptive PI controller eliminates the oscillations due to deviated parameters and operating conditions. The maximum oscillation amplitude is reduced to 400 mV from 2 V at the reference voltage changing from 25 V to 28 V.

**Author Contributions:** Conceptualization, Y.W. (Yinyu Wang) and B.H.; data curation, Y.W. (Yinyu Wang) and H.X.; formal analysis, Y.W. (Yinyu Wang) and D.Z.; investigation, Q.T.; methodology, B.H.; project administration, Y.W. (Yinyu Wang) and Q.T.; resources, Q.T.; software, Y.W. (Yuanxun Wang) and D.Z.; supervision, H.X.; validation, Y.W. (Yuanxun Wang) and H.X.; visualization, B.H. and Y.W.; writing—original draft, Y.W. (Yinyu Wang); writing—review and editing, D.Z and Q.T. All authors have read and agreed to the published version of the manuscript.

**Funding:** This research received no external funding.

**Data Availability Statement:** Data is contained within the article.

**Conflicts of Interest:** The authors declare no conflicts of interest.

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
