# Peer review of "A Transient Event-Capturing Circuit and Adaptive PI Control for a Voltage Mode Superbuck Converter"

_electronics, doi:10.3390/electronics13010107_

Round 1

Reviewer 1 Report

Comments and Suggestions for Authors

In this paper control of superbuck is addressed. But few issues are not clear:

1. We can measure the transient on digital oscilloscope. Why to use such a complicated circuit for same?

2. If topology is similar to buck, it may not have RHP zeroes. Please provide averaged state-space model and then obtain small signal transfer function. Use it to obtain transfer function and show RHP zeroes.

3. What are advantages of superbuck over buck.

4. In Fig. 3, I could not find the reference voltage for converter.

5. Please add results for load and line change.

6. You may compare the simulation oe experimental results with other advanced controllers.

Comments on the Quality of English Language

Improve grammer especially in abstract and introduction

Reviewer 2 Report

Comments and Suggestions for Authors

I read with pleasure the paper called "Transient Event Capturing Circuit and Adaptive PI Control for Voltage Mode Superbuck Converter"

I find the study interesting and worthy of publication.

I present the changes I deem necessary.

1) I wonder if capital letters are necessary in the title

2) in the abstract I would insert that an experimental evaluation was carried out with a power system of around 100 W.

3) I like the introduction, it is very direct and immediately addresses the issues. Maybe I would put what other researchers have done to address similar problems with divergent poles.

4) Figure 1 is intuitive, but having to write the Kirchhoff equations I would put some more references for voltage polarity and current directions.

5) I would also define D, for those less familiar with the work

6) I would define D hat better

7) line 111 correct the subscripts

8) fix equation 6 better

9) in figure 4, I would highlight the changes with other colors and I would put a better description in the legend

10) figure 5 is too qualitative, it could be improved

11) line 220 is not reactance, but impedance (imaginary)

12) figure 7 a, why red?

13) in 25k rad/s simulations, check

14) in the experimental part, I would immediately define the power size of the system

15) I would estimate the losses

16) conclusions should be more specific

Reviewer 3 Report

Comments and Suggestions for Authors

-In what ways does the transient event capture circuit help the superbuck converter monitor and improve its transient response?

-Which particular parts of the coupling and capturing circuits are involvedin the circuit?

-It is not clear how the up-crossing and down-crossing timings of the measured voltage are used by the capturing circuit to categorize various transient events?

-How does the transient responsiveness and stability of the superbuck converter get improved by the adaptive PI controller using the data from the transient event collecting circuit? 

- More dynamic results are needed for the controller validation.

Comments on the Quality of English Language

 Moderate editing of the English language is required in the introduction section.

Round 2

Reviewer 1 Report

Comments and Suggestions for Authors

Authors responded well.

Please add all ON state equations, OFF state equations in main paper and then show averaged model.

Please add block diagram of application with voltage level of each block.
